# Rheumatologist’s Perspective on Non-Infectious Uveitis: Patterns from Tertiary Referral Rheumatologic Clinics in Italy

**DOI:** 10.3390/ijms24119690

**Published:** 2023-06-02

**Authors:** Paola Triggianese, Mauro Fatica, Francesco Caso, Luisa Costa, Arianna D’Antonio, Marco Tasso, Elisabetta Greco, Paola Conigliaro, Alberto Bergamini, Claudia Fabiani, Luca Cantarini, Maria Sole Chimenti

**Affiliations:** 1Rheumatology, Allergology and Clinical Immunology, Department of “Medicina dei Sistemi”, PhD in Immunology, Molecular Medicine and Applied Biotechnology, University of Rome Tor Vergata, 00133 Rome, Italy; 2Department of Clinical Medicine and Surgery, School of Medicine and Surgery, University of Naples Federico II, 80131 Naples, Italy; 3Ophthalmology Unit, Department of Medicine, Surgery and Neurosciences, University of Siena, 53100 Siena, Italy; 4Department of Medical Sciences, Surgery and Neurosciences, Research Center of Systemic Autoinflammatory Diseases and Behcet’s Disease Clinic, University of Siena, 53100 Siena, Italy

**Keywords:** non-infectious uveitis, spondyloarthritis, Behçet disease, disease-modifying antirheumatic drugs, HLA-B27, HLA-B51

## Abstract

Non-infectious uveitis (NIU) can be an early or even the first extra-articular manifestation of systemic rheumatic diseases, or the first one; thus, rheumatologists are often involved in the diagnostic and therapeutic assessment of NIU. We evaluated 130 patients with a diagnosis of NIU who were admitted to two Italian rheumatologic clinics (Tor Vergata University Hospital in Rome, and Federico II University in Naples) from January 2018 to December 2021. Anterior uveitis (AU) occurred in 75.4% of patients, followed by posterior uveitis (PU, 21.5%); acute (54.6%) and recurrent (35.4%) NIU were more documented than chronic NIU (10%), and a bilateral involvement was observed in 38.7% of cases. Half of NIU cases were associated with spondyloarthritis (SpA); the remaining were affected by Behçet disease (BD)-related uveitis (13.9%) and idiopathic NIU (9.2%). HLA-B27^+^ patients (34.8%) had a higher prevalence of anterior and unilateral NIU (*p* = 0.005) with acute course (*p* = 0.04) than HLA-B27^–^ patients. On the contrary, HLA-B51^+^ patients (19.6%) had mostly PU and bilateral NIU (*p* < 0.0001) and recurrent course (*p* = 0.04) than HLA-B51^–^ patients. At the first rheumatologic referral, 117 patients (90%) received systemic treatments. Findings from this study demonstrate that rheumatologic referral has a pivotal role in the diagnostic work-up of NIU and may dramatically influence NIU-treatment strategies.

## 1. Introduction

Non-infectious uveitis (NIU) includes a heterogeneous group of diseases characterized by inflammation of the uvea that can cause significant ocular impairment up to blindness [1,2,3,4,5,6]. NIU shows a prevalence of about 30% in patients with spondyloarthritis (SpA), sarcoidosis, and Behçet disease (BD) [1]. According to the Standardization of Uveitis Nomenclature (SUN) Working Group, uveitis can be acute (≤3 months), chronic (>3 months), or recurrent [2,7,8]. Moreover, the anatomical classification categorizes uveitis into four groups, comprising anterior uveitis (AU), intermediate uveitis (IU), posterior uveitis (PU), and panuveitis (PanU). Among them, AU is the most common type (40.6%), followed by PanU (31.7%), PU (18.8%), and IU (9%) [9]. NIU is the most common non-articular clinical manifestation of SpA. [1,10,11]. The main symptoms of NIU include eye pain, ocular redness, blurred vision, floaters, and photophobia [10]. The prevalence of NIU ranges from 13% in undifferentiated SpA to 40% in axial SpA (ax-SpA) [2]. NIU affects about 37% of patients with inflammatory-bowel-disease SpA (IBD-SpA) and between 2% and 25% of patients with psoriatic arthritis (PsA) [12]. In this context, the presence of NIU, especially in young people with high C-reactive protein (CRP) levels and inflammatory low-back pain or peripheral arthritis, represents an additional element supporting the diagnostic hypothesis of SpA [13]. HLA-B27 could be revealed in up to 40% cases of NIU related to SpA [14,15,16]. Among patients with ax-SpA, NIU is more common in HLA-B27-positive than HLA-B27-negative patients, and thus, HLA-B27 positivity is associated with more frequent relapses and a worse prognosis [17]. As described, HLA-B27-associated NIU presents as acute AU, mainly unilateral, with an overall good prognosis. A chronic course or a high rate of recurrences increases the risk of complications [18,19]. On the other hand, BD-related uveitis presents often as a severe bilateral PanU with hypopyon, retinal necrotizing vasculitis, and a poor prognosis with a relatively high risk of blindness [20]. NIU is also the most frequent extra-articular involvement in juvenile idiopathic arthritis (JIA) [21]. The most common phenotypic presentation in JIA patients is a chronic anterior uveitis (CAU), which develops in 10–20% of cases and is usually pauci-symptomatic. Acute anterior uveitis (AAU) with onset in pediatric age occurs more frequently in HLA-B27-positive patients, especially in those with SpA, and is usually symptomatic and similar to that seen in adults [22,23]. NIU and its potential sight-threatening sequelae are, thus, responsible for significant disease-related morbidity in rheumatologic diseases. Furthermore, the adequate therapeutic choice of specific disease-modifying antirheumatic drugs (DMARDs), either conventional (cDMARDs) or biological (bDMARDs), should include NIU as a treatment target. The type of therapy for NIU depends on the anatomical location, laterality, and severity. Generally, in mild AU, therapy is based on topical corticosteroids (CCs) in association with mydriatic/cycloplegics drops [13]. Conversely, systemic CCs are recommended for the treatment of the acute phase, especially in more severe forms of non-anterior NIU [21]. Systemic CC sparing agents such as cDMARDs or bDMARDs are employed to control intraocular inflammation in the long term. During pediatric age, highly recurrent or chronic forms of AU and non-anterior uveitis can benefit from systemic immunosuppressants to spare systemic CCs and to avoid sight-threatening ocular complications such as cataract, glaucoma, band keratopathy, and macular edema [22]. Currently, among cDMARDs available for the treatment of NIU, methotrexate (MTX) has shown a higher response rate than mycophenolate mofetil (MMF), whereas cyclosporin (CsA) and azathioprine are comparably used for NIU in BD [24,25]. Among bDMARDs, monoclonal tumor-necrosis-factor inhibitors (TNFi) are indicated in non-anterior NIU and in pediatric CAU associated with JIA [26]. Infliximab (IFX) is reported to be successful in treating BD-related NIU and has been demonstrated to reduce the incidence of recurrences and improve the visual acuity of patients [27,28]. Finally, ongoing clinical trials are evaluating Janus-associated kinase inhibitors (JAKi) and phosphodiesterase inhibitors in NIU patients, with promising results [29,30,31,32].

Rheumatologists are often involved in the diagnostic and therapeutic assessment of patients with NIU, as NIU itself can be an early or even the first extra-articular manifestation of systemic rheumatic diseases. Given the relevant impact of NIU in rheumatologic patients [33], we aimed at describing clinical and therapeutic patterns in cohorts of patients with NIU from a population-based study carried out in tertiary-referral rheumatologic clinics in Italy in order to provide evidence from a rheumatologist’s perspective of both diagnosis and treatment management.

## 2. Results

We included 130 patients for a total of 179 affected eyes, with females representing 60% of the study population (*n* = 79). Data from the study cohort are summarized in Table 1. A previous or current history of smoking occurred in 33% (43/130) of patients, with a prevalent male distribution (*p* < 0.0001). Concomitant ophthalmologic diseases were revealed in 27.7% of eyes studied, mainly including episcleritis (25%) and keratoconjunctivitis sicca (16.7%). No differences resulted between males and females in age at the first rheumatologic referral (males, 43 ± 13.4 y.o. vs. females, 45.3 ± 12.9 y.o.) or age at NIU onset (males, 38.6 ± 13.8 y.o. vs. females 41.4 ± 13.5 y.o.). The referral to rheumatologists (Table 2) was prescribed by family doctors in half of the cohort (*n* = 59, 45.4%); in the remaining cases, patients were mainly referred by gastroenterologists (*n* = 31, 23.9%) and ophthalmologists (*n* = 28, 21.8%). Joint pain was the main reason for referral (*n* = 88, 67.7%), whereas NIU itself represented 27.7% of cases, without differences in sex (Table 2). Among the latter group of NIU patients, 72.2% of patients (*n* = 26) were referred to a rheumatologic evaluation by an ophthalmologist, and in 64% of them a diagnosis of SpA occurred.

### 2.1. Patterns of NIU in the Study Cohort

The most prevalent anatomic involvement was AU, which occurred in 75.4% of patients, followed by PU (21.5%); there were three cases of PanU and a single case of IU (Figure 1A). Half of the cohort (*n* = 71, 54.6%) experienced acute NIU, whereas the remaining had mainly recurrent NIU (*n* = 46, 35.4%), and chronic NIU was documented in 10% (*n* = 13) of cases (Figure 1B). The mean age at NIU onset was younger in patients with recurrent and chronic NIU than acute NIU (38 ± 12.3 y.o. vs. 42.6 ± 14.4 y.o., *p* = 0.02), with no differences in sex distribution. Unilateral NIU was registered in 81 patients (62.3%), with no difference in sex (Figure 1C). The age at NIU onset was older in patients with unilateral NIU than in patients with bilateral NIU (42.5 ± 14 y.o. vs. 36.36 ± 14 y.o., *p* = 0.01). Moreover, patients with unilateral-NIU had AU (*p* < 0.0005) and acute course (*p* < 0.00001) at a higher rate compared to patients with bilateral NIU (Figure 2).

### 2.2. Concomitant Rheumatologic Diseases

Almost a half of NIU cases had a diagnosis of SpA, including IBD-SpA (*n* = 25, 19.2%, 7 males and 18 females), ax-SpA (*n* = 23, 17.7%, 15 males and 8 females), and PsA (*n* = 22, 16.9%, 8 males and 15 females). BD occurred in 18 patients (13.9%, 11 males and 7 females) and RA in 11 patients (8.5%, 4 males and 7 females). NIU associated with other immune-mediated diseases such as connective-tissue diseases (CTDs) and small-vessel vasculitis was registered in 14 patients (10.8%), and JIA in five patients (3.9%). In 9.2% of subjects, NIU was defined as idiopathic (*n* = 12, 2 males and 10 females) (Table 3 and Figure 3). 

A higher prevalence of AU was found when comparing IBD-SpA (96% vs. 69.5%, *p* = 0.01), ax-SpA (95.7% vs. 71%, *p* = 0.02), and PsA (96% vs. 71.3%, *p* = 0.02) with patients without the respective type of arthritis. Moreover, patients with PsA had mostly acute-course (73.9% vs. 49.1%, *p* = 0.01) and unilateral NIU (78.3% vs. 58.3%, *p* = 0.03) compared with patients without PsA (Figure 4). Similarly, patients with IBD-SpA and ax-SpA had mostly acute and unilateral NIU compared to patients without the respective type of arthritis, but without statistically significative differences (Figure 5 and Figure 6, respectively). In patients with BD, a significantly higher prevalence of male sex (61.1% vs. 35.7%, *p* = 0.04), PU (88.9% vs. 12.5%, *p* < 0.00001), and bilateral NIU was recorded (77.8% vs. 31.3%, *p* = 0.0001) compared with patients without BD (Figure 7). In addition, the recurrent course showed a trend of being more frequent than acute and chronic in BD vs. non-BD subjects (*p* = 0.05). RA patients experiencing NIU (*n* = 11) showed mainly acute course (63.6%), anterior involvement (81.8%), and a unilateral pattern (63.6%). Similarly, CTD-associated NIU (*n* = 10) was more frequently acute (60%) and unilateral (80%), with anterior and posterior involvement equally distributed (50%). Small-vessel-vasculitis-associated NIU (*n* = 4) was more frequent in PU (75%), with a homogeneous distribution in terms of course and laterality. Among patients with idiopathic NIU (*n* = 12), AU was the most predominant anatomic involvement (*n* = 8, 66.7%), followed by two cases of PU and single cases of IU and PanU. These patients also showed more frequent bilateral involvement (*n* = 8, 66.7%). In terms of course, acute and recurrent idiopathic NIU were present in the same proportion (*n* = 5, 41.7%), with only two cases of chronic idiopathic NIU.

### 2.3. HLA Typing

A relevant proportion of NIU patients from the cohort (92/130, 70.7%) underwent an HLA-typing study for the detection of B27/B51 haplotypes. Overall, 32 patients were positive for HLA-B27 (34.8%) and 18 patients for HLA-B51 (19.6%). HLA-B27^+^ patients more frequently had AU (90.6% vs. 63.3%, *p* = 0.005), acute course (68.8% vs. 46.7%, *p* = 0.04), and unilateral involvement (75% vs. 55%, *p* = 0.05) than HLA-B27^–^ patients (Figure 8A). HLA-B27^+^ NIU had a significantly more prevalent diagnosis of ax-SpA than HLA-B27^–^ NIU (50% vs. 8.3%, *p* < 0.00001) than HLA-B27^–^ NIU. On the contrary, HLA-B51^+^ patients had mostly PU (61.1% vs. 13.5%, *p* < 0.0001), recurrent/chronic course (66.7% vs. 40.5%, *p* = 0.04), and bilateral involvement (77.8% vs. 28.4%, *p* = 0.0001) compared to HLA-B51^–^ patients (Figure 8B). A diagnosis of BD was performed more frequently in HLA-B51^+^ than in HLA-B51^–^ patients (77.8% vs. 5.4%, *p* < 0.00001). The age of NIU onset was younger in HLA-B51^+^ than in HLA-B51^–^ patients (34.2 ± 13.9 vs. 41.7 ± 13 y.o., *p* = 0.003). 

### 2.4. Treatment Strategies in NIU

Ninety percent of patients (*n* = 117) received a systemic treatment at the first rheumatologic referral (Table 4). Among them, the most frequent concomitant rheumatologic diseases were IBD-SpA (*n* = 23), ax-SpA (*n* = 22), PsA (*n* = 18), BD (*n* = 16), and RA (*n* = 11), whereas only nine patients had idiopathic NIU (Figure 9). CCs were administered in 37 patients (28.5%) as monotherapies, whereas 42.3% (55/130) of patients started cDMARDs as a first-line CC-sparing therapy. Most patients received sulfasalazine (SSZ, 32.7%) and MTX (29.1%). cDMARDs were administered in 22 patients (40%) in combination with CCs and in 19 patients (34.6%) as a monotherapy, whereas 9 patients received a combination therapy with a bDMARD (16.4%). A quarter of the cohort (25.4%, 33/130) started bDMARDs as a first-line DMARD therapy after the rheumatologic evaluation. The most prescribed bDMARD was adalimumab (12/33, 36.4%), followed by golimumab (5/33, 15.2%) and infliximab (4/33, 12.1%). The remaining cases were treated with etanercept (3/33, 9.1%), abatacept (2/33, 6.1%), secukinumab (2/33, 6.1%), rituximab (2/33, 6.1%), tocilizumab (1/33, 3%), certolizumab pegol (1/33, 3%), and ustekinumab (1/33, 3%). The bDMARDs were administered in combination with CCs in 7 cases (5.4%), whereas they were started as a monotherapy or combined with cDMARDs in 12 (9.2%) and 14 (10.8%) patients, respectively. Only a small percentage of patients (10%, *n* = 13) did not receive treatment at the first rheumatologic visit. In this group, 10 patients (76.9%) received a defined rheumatologic diagnosis during the follow-up, whereas the remaining three (23.1%) were diagnosed with idiopathic NIU (Figure 10). Among the latter three patients, two cases had bilateral AU and one case had bilateral parsplanitis, and no systemic treatment was indicated. At NIU onset, most patients were not on systemic treatment (*n* = 96, 73.9%), whereas 31 of them were on DMARDs, with 20 receiving cDMARDs and 11 bDMARDs (81.8% TNFi). Among cDMARD-treated patients, 55% of cases had a diagnosis of SpA, whereas BD and RA occurred in 20% and 15%, respectively. A prevalent diagnosis of SpA also occurred among bDMARD-treated patients (*n* = 8, 72.7%), whereas the remaining had JIA (*n* = 3, 27.3%). At NIU onset, 3/130 (2.3%) patients with AU were on systemic CCs in monotherapy. NIU was treated with topical CCs and cycloplegics in 75 patients (57.7%) and additional systemic CCs were employed in 55 patients (42.3%).

## 3. Discussion

Findings from this population-based study described the extent and patterns of NIU from real-life data by Italian adult patients referred to two tertiary rheumatologic clinics. Our data demonstrate that rheumatologic referral has a pivotal role in the diagnostic work-up of NIU and may dramatically impact treatment strategies in NIU. Particularly among patients referred to rheumatologists, NIU itself amounted to almost a third of the reasons for referral, with joint pain being the most frequent. NIU patients showed mainly AU, unilateral involvement, and acute course. Moreover, in half of them, a diagnosis of SpA, also including PsA, occurred. HLA-B27 and HLA-B51 positivity appeared to be related to different predominant NIU patterns. Furthermore, at the first rheumatologic referral, the majority of NIU patients received DMARD treatment. The present representative NIU cohort showed that both the age at onset and the sex distribution were comparable with data documented by the authors in recent Italian and Portuguese studies on NIU patients referred to rheumatology units [34,35]. Furthermore, we registered a slightly higher prevalence of females with NIU referred to rheumatologic units. Since immune-mediated diseases represent the main cause of NIU, it is fair to expect that NIU may predominantly affect females. A previous or current history of smoking occurred in a third of the cohort, with a prevalent male distribution: As is known, smoking may worsen the course of NIU, but having a previous or current smoking habit was not associated with any NIU pattern in our cohort [36]. Accordingly, the age at both NIU onset and first rheumatologic referral was similar between males and females. The rheumatologic referral was mainly addressed by family doctors, whereas a similar distribution occurred in the proportion of NIU patients referred by gastroenterologists and ophthalmologists. Although joint pain represented the main cause for referral, in up to a third of cases NIU itself represented the reason for the rheumatologic evaluation. Specifically, the latter patients were referred mainly by ophthalmologists, and in more than half of them, concomitant SpA or BD was diagnosed. These findings are consistent with evidence describing NIU as an early or even the first manifestation of such rheumatologic diseases and support the crucial role of ophthalmologists in promptly referring NIU patients to rheumatologists [37].

In our cohort, and in accordance with evidence from the literature, AU was the most prevalent anatomic involvement registered [9]. In addition, AU has been confirmed to be more frequent than PU in patients suffering from PsA or IBD-SpA [38,39,40,41]. Although IU is not very frequent, accounting for only about 10–15% of all NIU, it is interesting to note that only one patient affected by bilateral idiopathic IU, namely, parsplanitis, was discovered in our cohort. This patient was referred by an ophthalmologist for a systemic work-up, and since there were no associated ocular complications such as macular oedema, a systemic therapy was not started. However, in most cases, IU takes on idiopathic forms or is secondary to multiple sclerosis or sarcoidosis, and patients may be referred to neurologists or pulmonologists rather than rheumatologists [42]. Interestingly, patients experiencing recurrent and chronic NIU were younger at onset than patients with acute NIU, as were patients with bilateral NIU compared to patients with unilateral NIU. Patients with unilateral NIU also showed acute course and anterior involvement more frequently than patients with bilateral NIU, as well as a significantly higher proportion of concomitant PsA. Our analysis confirmed correlation between anterior involvement, unilaterality, and acute course, especially in PsA patients, as also highlighted in recent Korean and Spanish multicenter studies [43,44]. Several studies have attempted to characterize uveitis among patients with PsA. In a study on 16 adult PsA patients, Paiva and colleagues found that an insidious onset of uveitis, chronic course, bilateral involvement, and posterior eye disease were more common in association with PsA and IBD compared to the typical HLA-B27-associated NIU noted in patients with ankylosing spondylitis (AS) [45]. Recent evidence in a cohort of PsA patients demonstrated a prevalence of NIU at 4.9% with acute course in all cases, as well as anterior (80%) and unilateral (80%) involvement [46]. In the latter study, 45% of the patients diagnosed with PsA and NIU had HLA-B27 positivity: NIU showed features belonging to HLA-B27^+^-associated NIU, such as acute course, anterior involvement, and unilateral involvement. In our cohort of PsA patients, HLA-B27 was found in 40.9%, and similar to the aforementioned study, all of these patients had acute NIU, with almost all experiencing unilateral and anterior involvement, thus confirming the hypothesis of a potential HLA-B27 signature in characterizing NIU clinical presentation in these patients. No differences in sex have been revealed in the distribution of patterns of NIU in the type of course, in the whole cohort, or by stratifying according to associated rheumatologic diseases. Regarding the impact of sex on NIU, current data suggest that acute AU associated with SpA affects males more frequently than females, and typically occurs in young adults [16]. Moreover, male sex, hypopyon, increased ESR, or an associated SpA are potential risk factors for frequent relapses of HLA-B27-associated NIU [47]. In addition, the authors reported that males were more likely than females to develop psoriasis or PsA among patients with NIU [48]. Regarding concomitant rheumatologic diagnoses, almost half of NIU patients had a diagnosis of IBD-SpA, ax-SpA, PsA, or BD, with a similar prevalence. RA and CTDs represented the minority of cases, since in these pathologies NIU occurs less frequently than other ocular involvements such as dry eye (keratoconjunctivitis sicca) or episcleritis [49]. Interestingly, in almost 10% of patients, a defined rheumatologic diagnosis could not be performed; thus, NIU was defined as idiopathic. In these patients, AU and bilateral involvement were registered more frequently. PsA patients as well as BD patients showed a more defined NIU pattern by reporting AU, acute involvement, and unilateral involvement in the first group (PsA) and PU, recurrent/chronic, and bilateral involvement in the second (BD). In addition, in patients with concomitant SpA (both axial SpA and IBD-SpA) the most frequent NIU was AU. These results in our cohort are consistent with those reported in the literature [50,51]. Given the close association between NIU and HLA [52], a relevant proportion of NIU patients from the cohort underwent an HLA-typing study for the detection of B27/B51 haplotypes, mainly at the follow-up and without a difference in sex. The HLA-typing study was requested especially in patients with signs and symptoms of SpA (such as inflammatory low-back pain and dactylitis) and BD (such as oral and genital ulcers and folliculitis), in the context of providing further elements in support of the diagnostic suspicion. The prevalence of HLA-B27 positivity was almost 35%, whereas that of HLA-B51 positivity amounted to 20%. Furthermore, our study confirmed that HLA-B27^+^ patients tended to develop an associated AU, which can be often acute or recurrent and unilateral. In fact, approximately 60 to 80% of these NIU patients are HLA-B27^+^, whereas SpA is detected in 58% to 78% of cases among HLA-B27^+^ NIU patients [53]. In our study cohort, a diagnosis of BD occurred more frequently in HLA-B51^+^ than in HLA-B51^–^ patients: Accordingly, HLA-B51^+^ patients had mostly PU, recurrent/chronic course, and bilateral involvement than HLA-B51^–^ patients. Interestingly, the age of NIU onset was younger in HLA-B51^+^ than in HLA-B51^–^ patients, and, since HLA-B51 positivity is correlated to a greater probability of developing a clinical phenotype of BD with more severe ocular and neurological involvement [54], it is reasonable to also have expected a more precocious onset of NIU than in HLA-B51^–^ patients. HLA genotyping was performed in all subjects using real-time polymerase chain reaction (PCR). Next-generation sequencing (NGS)-based HLA typing could be performed in future investigations since HLA typing by NGS seems to provide superior accuracy compared to traditional methods, in accordance with evidence from recent studies [55].

Uveitis can also occur in RA, CTDs, and vasculitis, but less frequently than other ocular involvements, such as scleritis, episcleritis, and keratoconjunctivitis sicca [49]. Nevertheless, in 25 NIU patients of our cohort (19.2%) it was possible to formulate a concomitant diagnosis of one of these diseases (11 RA, 10 CTDs, 4 small-vessels vasculitis). AU was the most frequent anatomic involvement and the most frequent course was acute in all these diseases, as reported in the literature [49], whereas unilaterality was more frequently documented only in RA and CTDs. From the whole NIU cohort, we identified a subset of patients, representing a quarter of the cohort, who were already on treatment at NIU onset, and they were mainly on cDMARDs. However, the proportion of patients receiving additional therapies at the rheumatological follow-up was similar in patients who were already on treatment at NIU onset and in those who were treatment-naive at NIU onset. At the first rheumatologic referral, almost half of patients started cDMARDs as a first-line CC-sparing therapy, in monotherapy or combination, whereas bDMARDs were used in a quarter of patients. The choice of treatment was made considering not only NIU itself but also any eventual associated rheumatologic disease, with the selection of a therapy that could control all clinical manifestations of the disease. Of note, in patients with active NIU or a history of relapsing NIU, a therapy aimed at controlling ocular inflammation was preferred, in accordance with evidence-based guidelines [56,57]. 

The present study documents how a multidisciplinary approach can improve the rate of appropriate diagnosis, giving patients prompt access to specific treatment. The NIU diagnostic work-up is closely related to prompt collaboration among different clinicians, mainly including ophthalmologists and rheumatologists. This is because the diagnosis of NIU can be challenging, and a multi-disciplinary approach is needed to improve the early detection and the targeted management of patients. Systemic diseases such as IBD, SpA, etc., can affect patients with NIU who concurrently present an increased risk of developing severe visual disturbance such as blindness or low vision. Representative associated diseases vary depending on the type of NIU, as BD is mainly associated with panuveitis, posterior NIU, and anterior NIU; sarcoidosis with posterior and intermediate NIU; and JIA, SpA, and IBD with anterior NIU. Panuveitis can be also related to other associated autoimmune disorders and Vogt–Koyanagi–Harada syndrome. Nevertheless, diagnosis of NIU can be a challenge because the cause is often not clinically evident, and the differential diagnosis depends on the recognition that uveitis is not a unique disease but can be considered a group of disorders based on its potential associations with primary ocular conditions as well as systemic inflammatory diseases [12]. Starting from these considerations, all patients with NIU should be considered to be affected by more than one disorder, including ocular diseases and the potential associated systemic disease. The combined approach with a tight follow-up, specifically a closer collaboration between ophthalmologists and rheumatologists, who play a major role in diagnosing and providing appropriate treatments for patients with systemic immune disease, may benefit long-term outcomes. Although corticosteroids remain the first drug of choice in the case of active uveitis, given the risk of developing both ocular and systemic side effects, it is advisable to use them only for short periods. Hence, especially for recurrent and chronic uveitis, there is a need to treat and control associated systemic disorders using steroid-sparing immunosuppressive drugs. In fact, a recent retrospective study showed that the combined ophthalmologic and rheumatologic approach led to more usage and variety of immunosuppressive drugs compared to independent management [58]. Certainly, there is a lack of prospective studies evaluating how the early multidisciplinary approach could better control ocular inflammation and reduce the development of complications and reduced visual acuity. How the combined approach affects the treatment strategies and outcomes in patients with NIU certainly requires further prospective investigations aiming at assessing both the clinical outcome and the response to therapy in order to provide more evidence on potential targets to treat in NIU by stratifying patients according to age, sex, and associated diseases. 

## 4. Materials and Methods

### 4.1. Patients

The entry-selection criterion of this cross-sectional descriptive study consisted of having a diagnosis of NIU delivered by an expert ophthalmologist. Additional inclusion criteria were (1) age of NIU onset ≥ 16 years old, and (2) admission to the rheumatologic clinics at Tor Vergata University Hospital in Rome and Federico II University Hospital in Naples (Italy) between January 2018 and December 2021. NIU clinical presentation was classified in accordance with the Standardization of Uveitis Nomenclature (SUN) Working Group criteria [59]. The demographic and clinical data collected were sex, smoking habits, age at onset and admission, course and anatomic patterns, HLA-B27 and -B51 presence/absence, reason for rheumatologic referral, associated rheumatologic diseases, and treatments. A diagnosis of concomitant SpA, SpA-IBD, PsA, BD, JIA, and rheumatoid arthritis (RA) was defined in accordance with the respective international classification criteria [60,61,62,63,64,65]. In all cases, HLA typing was performed on patients’ DNA that was extracted from venous EDTA-anticoagulated blood and analyzed by using real-time polymerase chain reaction (PCR) according to the manufacturer’s instructions (XeliGen RT System, Eurospital SPA, Italy, in most cases, and analogous RealTime PCR-based systems in the remaining cases).

### 4.2. Statistical Analysis

Mean and standard deviation (SD) expressed normally distributed variables. Continuous variables were compared using the parametric unpaired t test or the nonparametric Mann–Whitney U test when appropriate. Categorical variables were presented with absolute frequencies and percentages and were compared using the Chi-squared test or Fisher’s exact test when appropriate. A *p*-value < 0.05 was considered significant. All statistical analyses were performed using GraphPad Prism version 9 (GraphPad Software, San Diego, CA, USA).

## 5. Conclusions

To the best of our knowledge, this is the first study that evaluated which type of physician refers patients with NIU to rheumatologic units, and it is important to note that in most cases family doctors play this pivotal role. Hence, there is a need to establish short routes that allow patients to be directed to reference centers directly from the territory, especially those who do not have active uveitis, so they can quickly access dedicated facilities such as ophthalmological emergency rooms. Our findings document that the interplay between ophthalmologists and rheumatologists is crucial in the setting of NIU and, above all, there should be an early rheumatological referral to diagnose any underlying rheumatological disease early and offer the patient a potential timely immunosuppressive treatment. Hence, there is a need to strengthen the role of international registries that help clinicians in caring for patients with NIU by collecting data on the course of disease and the response to therapies [66]. Prospective observational registries evaluating clinical characteristics, management, and outcomes can facilitate clinicians in their complex decision-making processes in NIU diagnostic and therapeutic work-up, and, thus, can improve the clinical care in the setting of a multicenter collaboration.

## Figures and Tables

**Figure 1 ijms-24-09690-f001:**
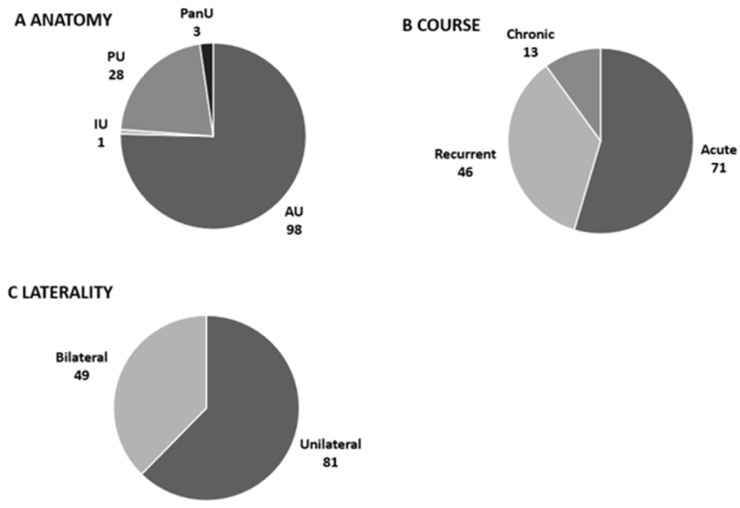
Patterns of noninfectious uveitis in the study cohort. Anatomic involvement (**A**), course (**B**), and laterality (**C**). NIU, noninfectious uveitis; AU, acute NIU; IU, intermediate NIU; PU, posterior NIU; PanU, panuveitis.

**Figure 2 ijms-24-09690-f002:**
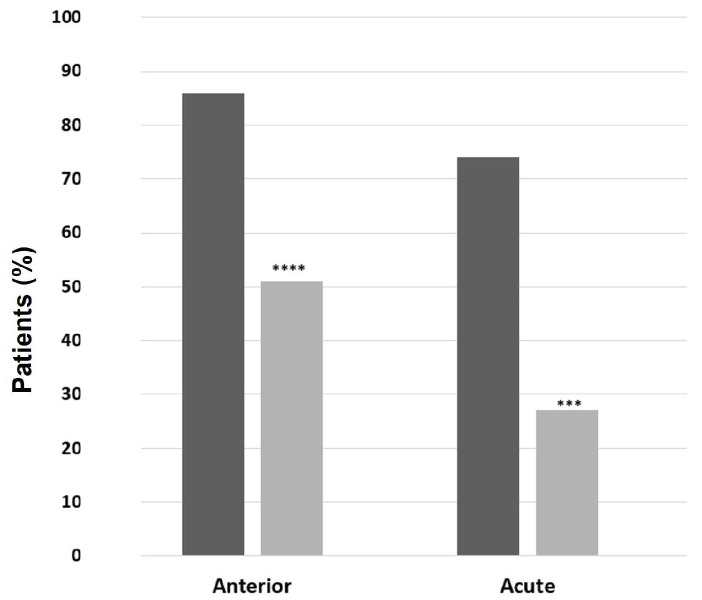
Patients with unilateral and bilateral noninfectious uveitis. In black, unilateral; in grey, bilateral. Categorical variables were presented with percentages and were compared using the Chi-squared test or Fisher’s exact test when appropriate. A *p*-value < 0.05 was considered significant (*** *p* < 0.001; **** *p* < 0.0001).

**Figure 3 ijms-24-09690-f003:**
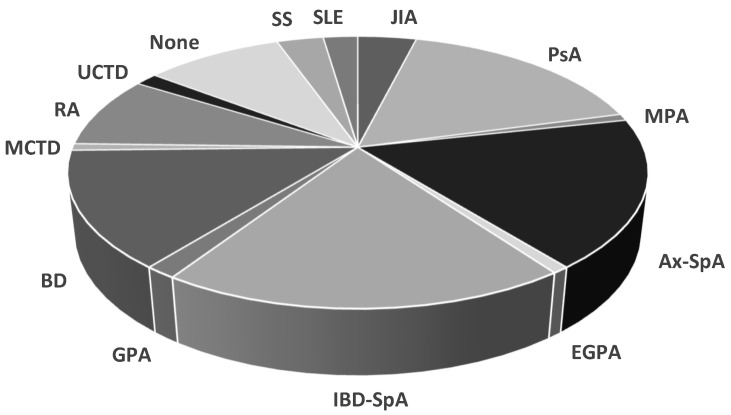
Rheumatologic diagnoses in the study cohort. Inflammatory-bowel-disease (IBD)-associated spondyloarthritis (SpA), axial spondyloarthritis (ax-SpA), psoriatic arthritis (PsA), Behçet disease (BD), rheumatoid arthritis (RA), juvenile idiopathic arthritis (JIA), Sjögren syndrome (SS), systemic lupus eritematosus (SLE), undifferentiated connective-tissue disease (UCTD), mixed connective-tissue disease (MCTD), granulomatosis with polyangiitis (GPA), eosinophilic granulomatosis with polyangiitis (EGPA), microscopic polyangiitis (MPA).

**Figure 4 ijms-24-09690-f004:**
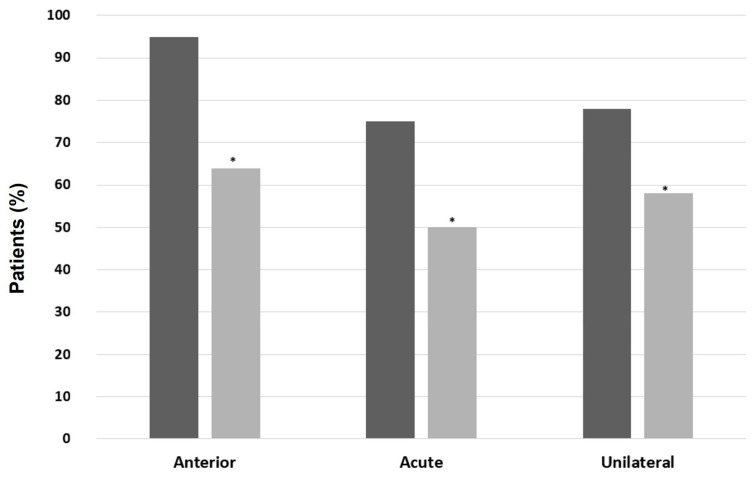
Patterns of noninfectious uveitis in patients with and without psoriatic arthritis (PsA). In black, patients with PsA; in grey, patients without PsA. Categorical variables were presented with percentages and were compared using the Chi-squared test or Fisher’s exact test when appropriate. A *p*-value < 0.05 was considered significant (* *p* < 0.05).

**Figure 5 ijms-24-09690-f005:**
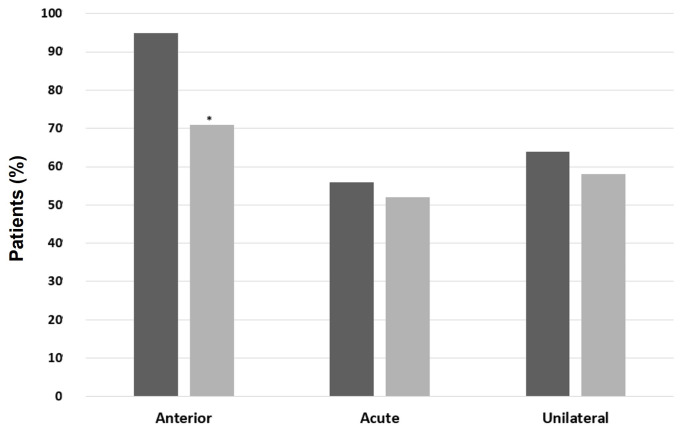
Patterns of noninfectious uveitis in patients with and without inflammatory-bowel-disease-associated spondyloarthritis (IBD-SpA). In black, patients with IBD-SpA; in grey, patients without IBD-SpA. Categorical variables were presented with percentages and were compared using the Chi-squared test or Fisher’s exact test when appropriate. A *p*-value < 0.05 was considered significant (* *p* < 0.05).

**Figure 6 ijms-24-09690-f006:**
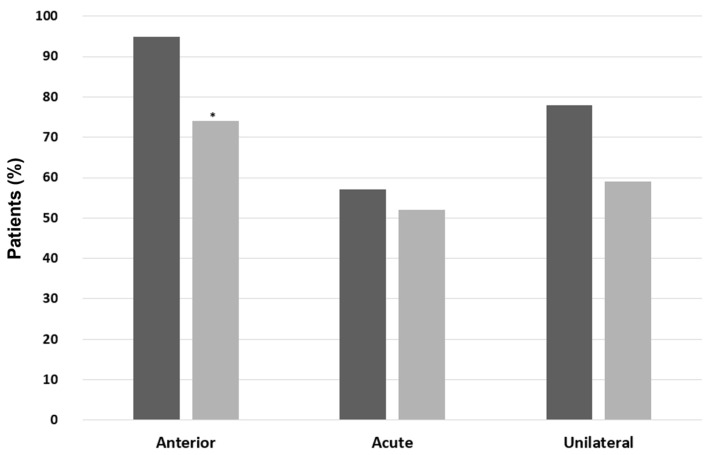
Patterns of noninfectious uveitis in patients with and without axial spondyloarthritis (ax-SpA). In black, patients with ax-SpA; in grey, patients without ax-SpA. Categorical variables were presented with percentages and were compared using the Chi-squared test or Fisher’s exact test when appropriate. A *p*-value < 0.05 was considered significant (* *p* < 0.05).

**Figure 7 ijms-24-09690-f007:**
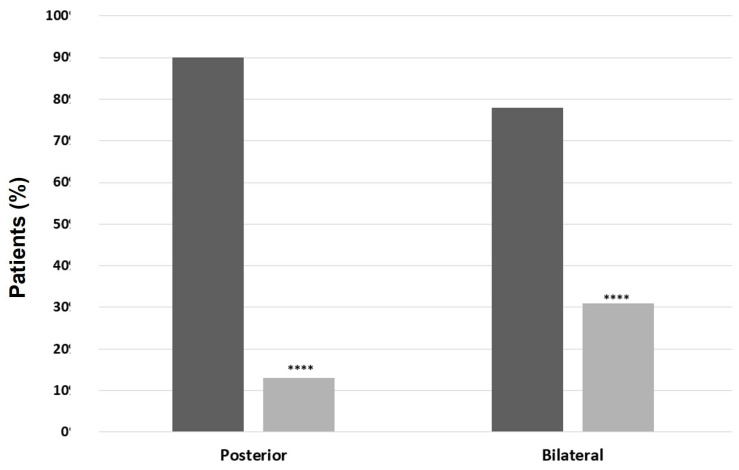
Patterns of noninfectious uveitis in patients with and without Behçet disease (BD). In black, patients with BD; in grey, patients without BD. Categorical variables were presented with percentages and were compared using the Chi-squared test or Fisher’s exact test when appropriate. A *p*-value < 0.05 was considered significant (**** *p* < 0.0001).

**Figure 8 ijms-24-09690-f008:**
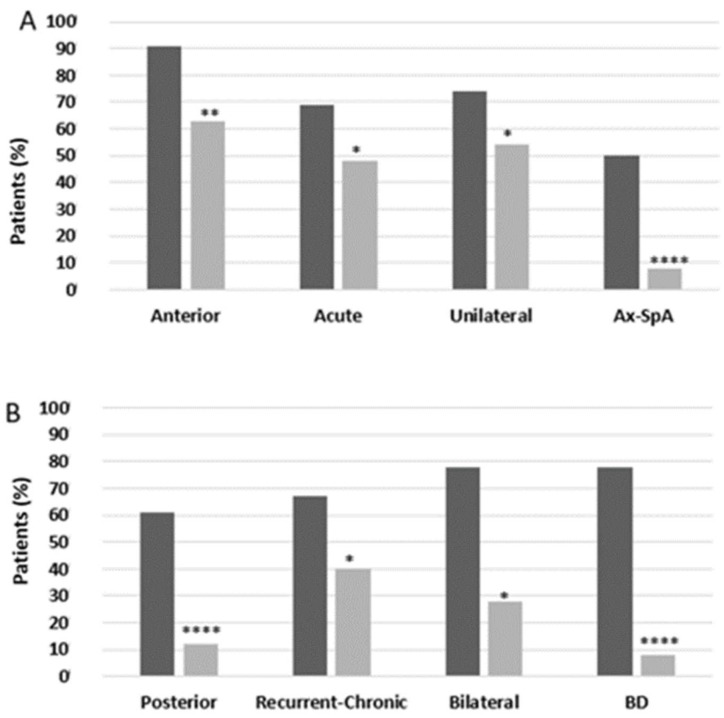
Patterns of noninfectious uveitis in accordance with human-leukocyte-antigen (HLA) distribution. (**A**) HLA-B27-positive (black) vs. HLA-B27-negative (grey) patients. (**B**) HLA-B51-positive (black) vs. HLA-B51-negative (grey) patients. Ax-SpA, axial spondyloarthritis; BD, Beçhet disease. Categorical variables were presented with percentages and were compared using the Chi-squared test or Fisher’s exact test when appropriate. A *p*-value < 0.05 was considered significant (* *p* < 0.05; ** *p* < 0.01; **** *p* < 0.0001).

**Figure 9 ijms-24-09690-f009:**
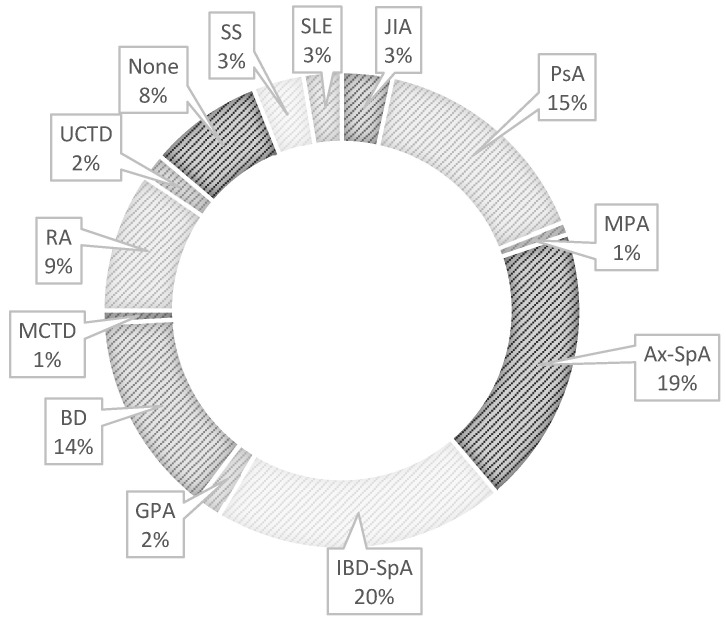
Concomitant rheumatologic diagnoses in patients with noninfectious uveitis receiving therapy at first rheumatologic referral. Inflammatory-bowel-disease-associated spondyloarthritis (IBD-SpA), axial spondyloarthritis (ax-SpA), psoriatic arthritis (PsA), Behçet disease (BD), rheumatoid arthritis (RA), juvenile idiopathic arthritis (JIA), Sjögren syndrome (SS), systemic lupus eritematosus (SLE), undifferentiated connective-tissue disease (UCTD), mixed connective-tissue disease (MCTD), granulomatosis with polyangiitis (GPA), microscopic polyangiitis (MPA).

**Figure 10 ijms-24-09690-f010:**
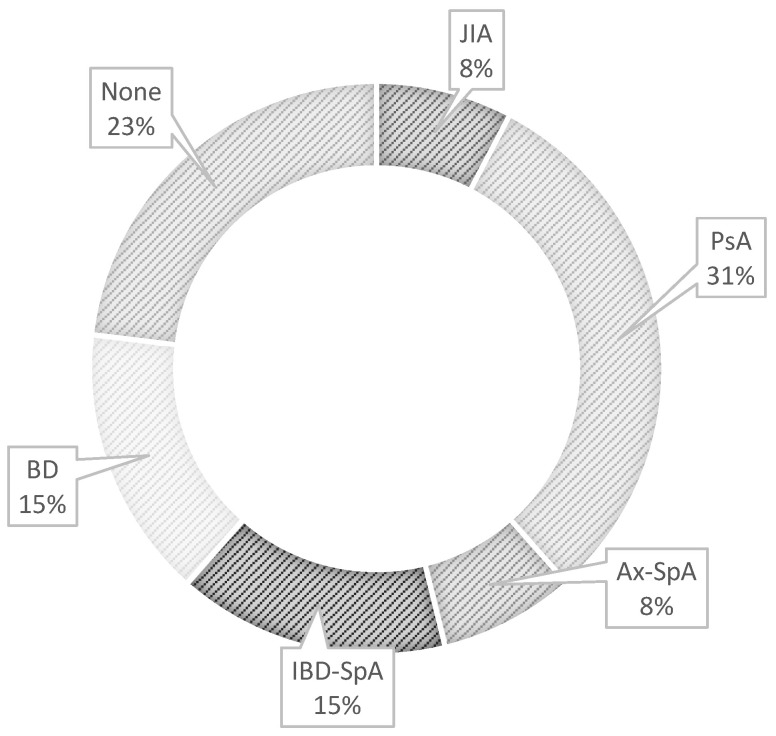
Concomitant rheumatologic diagnoses in patients with noninfectious uveitis non receiving therapy at first rheumatologic referral. Inflammatory-bowel-disease-associated spondyloarthritis (IBD-SpA), axial spondyloarthritis (ax-SpA), psoriatic arthritis (PsA), Behçet disease (BD), juvenile idiopathic arthritis (JIA).

**Table 1 ijms-24-09690-t001:** Demographic data from patients in the study populations.

	All	Males	Females
Patients; *n*	130	51	79
Age at referral (years); mean ± SD (min–max)	44.9 ± 13.4 (16–70)	43 ± 13.4 (17–70)	45.3 ± 12.9 (16–70)
Age at NIU onset (years); mean ± SD (min–max)	40.4 ± 13.9 (16–68)	38.6 ± 13.8 (16–66)	41.4 ± 13.5 (16–68)
Referral delay (months); mean ± SD	63.1 ± 44.6	58.2 ± 40.8	66.3 ± 42.5
Smoking history; *n* (%)	43 (33.1)	27 (52.9) *	16 (20.3)
Other ophthalmologic diseases; *n* eyes (%)	36 (27.7%)	12 (23.5)	24 (30.4)
Episcleritis; *n* (%)	9 (6.9)	4 (7.8)	5 (6.3)
Keratoconjunctivitis sicca; *n* (%)	6 (4.6)	1 (2)	5 (6.3)
Age-related maculopathy; *n* (%)	3 (2.3)	2 (3.9)	1 (1.3)
Cataract; *n* (%) Ɨ	3 (2.3)	0 (0)	3 (3.8)
Glaucoma; *n* (%) Ɨ	3 (2.3)	0 (0)	3 (3.8)
Retinopathy of other causes; *n* (%) ƗƗ	2 (1.5)	0 (0)	2 (2.5)
Thrombosis of retinal vessels; *n* (%)	2 (1.5)	1 (2)	1 (1.3)
Retinal vasculitis; *n* (%)	1 (0.8)	1 (2)	0 (0)
Peripheral ulcerative keratitis; *n* (%)	1 (0.8)	0 (0)	1 (1.3)

NIU; noninfectious uveitis; SD, standard deviation; * *p* = 0.0001. Ɨ all findings of cataracts and glaucoma were diagnosed prior to NIU onset; ƗƗ diabetic and hypertensive retinopathy was found in two eyes of two different female patients.

**Table 2 ijms-24-09690-t002:** Rheumatologic referral.

Reason for Rheumatologic Referral	N (%) (F:M)
Joint pain	88 (67.7) (1.4:1)
Noninfectious uveitis	36 (27.7) (1.4:1)
Fever of unknown origin	2 (1.5) (1:1)
Urticaria–angioedema	2 (1.6) (1:1)
Interstitial lung disease	1 (0.8) (1:1)
Pericarditis	1 (0.8) (1:1)
**Referred by**	**N (%)**
Family doctor	59 (45.4)
Gastroenterologist	31 (23.9)
Ophthalmologist	28 (21.5)
Dermatologist	3 (2.3)
Infectious-disease specialist	3 (2.3)
Neurologist	3 (2.3)
Pulmonologist	2 (1.5)
Cardiologist	1 (0.8)

**Table 3 ijms-24-09690-t003:** Rheumatologic diagnoses in the study cohort.

Rheumatologic Diagnosis	N (%)
Inflammatory-bowel-disease-associated spondyloarthritis (IBD-SpA)	25 (19.2)
Axial spondyloarthritis (ax-SpA)	23 (17.7)
Psoriatic arthritis (PsA)	22 (16.9)
Behçet disease (BD)	18 (13.9)
Rheumatoid arthritis (RA)	11 (8.5)
Juvenile idiopathic arthritis (JIA)	5 (3.9)
Sjögren syndrome (SS)	4 (3.1)
Systemic lupus eritematosus (SLE)	3 (2.3)
Undifferentiated connective-tissue disease (UCTD)	2 (1.5)
Mixed connective-tissue disease (MCTD)	1 (0.8)
Granulomatosis with polyangiitis (GPA)	2 (1.5)
Eosinophilic granulomatosis with polyangiitis (EGPA)	1 (0.8)
Microscopic polyangiitis (MPA)	1 (0.8)
None	12 (9.2)

**Table 4 ijms-24-09690-t004:** Treatment strategies at first rheumatological referral.

Type of Systemic Therapy	N (%)
CCs	37 (28.5)
cDMARDs	19 (14.6)
cDMARDs + CCs	22 (16.9)
bDMARDs	12 (9.3)
bDMARDs + CCs	7 (5.4)
bDMARDs + cDMARDs	9 (6.9)
bDMARDs + cDMARDs + CCs	5 (3.8)
None	19 (14.6)
**DMARD Drug Administered**	**N (%)**
Methotrexate	23 (17.7)
Sulfasalazine	26 (20.0)
Hydroxycloroquine	7 (5.4)
Leflunomide	4 (3.1)
Azathioprine	4 (3.1)
Cyclophosphamide	2 (1.5)
Colchicine	6 (4.6)
Cyclosporine A	4 (3.1)
Adalimumab	12 (9.3)
Certolizumab	1 (0.8)
Infliximab	5 (3.8)
Golimumab	6 (4.6)
Etanercept	4 (3.1)
Rituximab	2 (1.5)
Secukinumab	2 (1.5)
Abatacept	2 (1.5)
Tocilizumab	1 (0.8)
Ustekinumab	1 (0.8)

CCs, corticosteroids; cDMARDs, conventional synthetic disease-modifying antirheumatic drugs; bDMARDs, biological disease-modifying antirheumatic drugs.

## Data Availability

The data from this study are available upon a reasonable request addressed to the corresponding author.

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
