# Peer review of "Rheumatologist’s Perspective on Non-Infectious Uveitis: Patterns from Tertiary Referral Rheumatologic Clinics in Italy"

_ijms, 2023, doi:10.3390/ijms24119690_

Round 1
Reviewer 1 Report
The manuscript submitted by Triggianese and colleagues presents an evaluation of Italian chort of NIU patients in regard to rheumatologic diagnosis, HLA-B27 and HLA-B51 typing and applied treatment. The study confirms previous findings in other cohorts and highlights the significance of rheumatologic referral of NIU patients and its role for subsequently prescribed therapy.
This manuscript needs thorough revision before publication.
Main points:
1) The concept of the study must be clearly defined and concordant in the abstract and the introduction;
2) The quality of the figures must be improved;
3) There is significant discrepancy between the results description and data presented in Table 2. The table shows that only 3 patiens were referred to rheumatologists by family doctors while the main message from the results section and the conclusions states that family doctors have a main role for rheumatologic referral. Please explain the reason for this discrepancy.
4) Include gender data in Table 2 in order to support the description of results and the statement for lack of gender differences (L108-109).
5) Figure data for concomitant rheumatologic diseases must be supplemented. Figure 3 and 4 may be combined and also, figures for NIU patterns in IBD-SpA and ax-SpA must be included;
6) There is no information regarding patient diagnosis and type of therapy applied at first rheumatologic referral. Please explain how the data from Table 4 support the conclusion L405-409.
7) A better discussion on NIU diagnostic work-up and its influence on NIU treatment strategies is needed.
8) Materials and methods: include information for HLA typing.
Minor points:
-Revise the title of Figure 2;
-Table 3: data for IBD-SpA is shown twice.
Author Response
Point-by-point reply to Reviewers
Reviewer 1.
The manuscript submitted by Triggianese and colleagues presents an evaluation of Italian cohort of NIU patients in regard to rheumatologic diagnosis, HLA-B27 and HLA-B51 typing and applied treatment. The study confirms previous findings in other cohorts and highlights the significance of rheumatologic referral of NIU patients and its role for subsequently prescribed therapy. This manuscript needs thorough revision before publication.
Main points:
R1. 1) The concept of the study must be clearly defined and concordant in the abstract and the introduction.
Authors. According to the Reviewer, the concept of the study has been defined in the introduction that has been also modified in order to make it more concordant with the abstract [lines 85-91], as suggested.
R1. 2) The quality of the figures must be improved.
Authors. As suggested, the revised manuscript included figures with a better quality.
R1. 3) There is significant discrepancy between the results description and data presented in Table 2. The table shows that only 3 patients were referred to rheumatologists by family doctors while the main message from the results section and the conclusions states that family doctors have a main role for rheumatologic referral. Please explain the reason for this discrepancy.
Authors. The revised manuscript included correct data (the discrepancy was due to an input error).
R1. 4) Include gender data in Table 2 in order to support the description of results and the statement for lack of gender differences (L108-109).
Authors. According to the Reviewer, the revised paper included gender data in Table 2.
R1. 5) Figure data for concomitant rheumatologic diseases must be supplemented. Figure 3 and 4 may be combined and also, figures for NIU patterns in IBD-SpA and ax-SpA must be included.
Authors. The paper has been modified in accordance with all the suggestions (concomitant rheumatologic diseases, figures 3 and 4, figures for NIU patterns in IBD-SpA and ax-SpA).
R1. 6) There is no information regarding patient diagnosis and type of therapy applied at first rheumatologic referral. Please explain how the data from Table 4 support the conclusion L405-409.
Authors. Patients’ diagnoses are reported in figure 4. Data reported in Table 4 describe that cDMARDs represent, in both monotherapy and combination, almost a half of used drugs (42.2%) while bDMARDs were administered in a quarter of patients (25.4%, including both monotherapy and combination therapy).
R1. 7) A better discussion on NIU diagnostic work-up and its influence on NIU treatment strategies is needed.
Authors. The revised manuscript included a better discussion on NIU diagnostic work-up and treatment strategies, according to the suggestions [lines 413-447].
R1. 8) Materials and methods: include information for HLA typing.
Authors. The revised manuscript better described that HLA genotyping was performed in all subjects by using real-time polymerase chain reaction (PCR) according to the manufacturer’s instructions.
R1. Minor points: Revise the title of Figure 2; Table 3: data for IBD-SpA is shown twice.
Authors: All the minor changes have been addressed in the revised manuscript.

Reviewer 2 Report
An interesting work summarizing the population of patients with uveitis. Interesting work, well described, presented and well discussed population. In principle, the work could be published, but to increase the quality, please respond to a few inquiries:
- in table 1 - characteristics of the population, the average age and deviation are given. It would be good if the age of the youngest and oldest patient was also displayed.
- in table 3 - treatment: not all drugs in the list "immunosuppressants" are really like that. I suggest either removing the title "immunosuppressants" or dividing them into therapeutic groups.
- have high-throughput genetic tests been performed?
- I wonder why cardiologists reported the most patients? Maybe it's worth discussing? this is important when planning education addressed to physicians of various specialties
Author Response
Reviewer 2.
An interesting work summarizing the population of patients with uveitis. Interesting work, well described, presented and well discussed population. In principle, the work could be published, but to increase the quality, please respond to a few inquiries:
R2. In table 1 - characteristics of the population, the average age and deviation are given. It would be good if the age of the youngest and oldest patient was also displayed.
Authors. The revised table 1 has been modified according to the Reviewer.
R2. In table 3 - treatment: not all drugs in the list "immunosuppressants" are really like that. I suggest either removing the title "immunosuppressants" or dividing them into therapeutic groups.
Authors. The revised manuscript addressed the suggested changes.
R2. Have high-throughput genetic tests been performed?
Authors. The revised manuscript better described that HLA genotyping was performed in all subjects by using real-time polymerase chain reaction (PCR) according to the manufacturer’s instructions. The Next-Generation Sequencing (NGS)-based HLA typing could be performed in future investigations since HLA typing by NGS seems to provide superior accuracy compared to traditional methods, in accordance with evidence from recent studies.
R2. I wonder why cardiologists reported the most patients. Maybe it's worth discussing? This is important when planning education addressed to physicians of various specialties
Authors. The revised manuscript included correct data (ʺcardiologistsʺ was due to an input error).

Round 2
Reviewer 1 Report
The manuscript has been substantially improved. However, the conclusions, especially the last sentence, rise a fundamental question - how could the course of the disease and response to systemic treatment be predicted based on the authors' findings and/or international registries? Please, explain or correct the conclusions.
The authors did not mark all changes they made in the text.
There are some technical issues that need to be addressed:
1) Asterisk signs on the figures should be properly positioned - either above the bar with significant decrease, or above (in the middle) the two bars that are compared.
2) The authors stated: "In all patients, the method for HLA genotyping was the real-time polymerase chain reaction (PCR), performed according to the manufacturer’s instructions."
Information about the manufacturer should be included.
Author Response
R1. Comments and Suggestions for Authors
The manuscript has been substantially improved. However, the conclusions, especially the last sentence, rise a fundamental question - how could the course of the disease and response to systemic treatment be predicted based on the authors' findings and/or international registries? Please, explain or correct the conclusions.
Authors. According to the Reviewer, the conclusions have been revised (lines 484-490).
R1. The authors did not mark all changes they made in the text.
Authors. In the revised manuscript, all changes in the text have been highlighted.
R1. There are some technical issues that need to be addressed: 1) Asterisk signs on the figures should be properly positioned - either above the bar with significant decrease, or above (in the middle) the two bars that are compared.
Authors. According to the Reviewer, asterisk signs on the figures have been properly positioned.
R1. 2) The authors stated: "In all patients, the method for HLA genotyping was the real-time polymerase chain reaction (PCR), performed according to the manufacturer’s instructions." Information about the manufacturer should be included.
Authors. According to the Reviewer, details on PCR systems have been included (lines 462-466).
